# Marker-Assisted Pyramiding of *CRa* and *CRd* Genes to Improve the Clubroot Resistance of *Brassica rapa*

**DOI:** 10.3390/genes13122414

**Published:** 2022-12-19

**Authors:** Xiaonan Li, Yingxia Wei, Yingmei Ma, Guizhu Cao, Siwen Ma, Tianyu Zhang, Zongxiang Zhan, Zhongyun Piao

**Affiliations:** College of Horticulture, Shenyang Agricultural University, Shenyang 110866, China

**Keywords:** clubroot, Chinese cabbage, gene pyramiding, *CRa*, *CRd*

## Abstract

Clubroot, caused by *Plasmodiophora brassicae*, is an economically important soil-borne disease that threatens Brassicaceae crops worldwide. In recent years, the incidence area of Chinese cabbage (*Brassica rapa* ssp. *pekinensis*) clubroot disease has increased, which severely affects the yield and quality of Chinese cabbage. The resistance of varieties harboring the single clubroot-resistance (CR) gene is easily broken through by *P. brassicae* pathotypes. *CRa* and *CRd*, genetically identified in *B. rapa*, are CR genes known to be highly resistant to different *P. brassicaea* pathotypes. In our study, we perform the gene pyramiding of *CRa* and *CRd* in Chinese cabbages through marker-assisted selection (MAS), and develop homozygous pyramided lines. The newly generated pyramided lines exhibit greater resistance to six different pathotypes than that of two parental lines carrying a single CR gene. This study provides new CR-gene-pyramided lines for the development of clubroot-resistant Brassica varieties for future breeding programs.

## 1. Introduction

Chinese cabbage (*B. rapa* L. ssp. *pekinensis*), belonging to the Brassica subspecies of the Brassicaceae family, is a type of leafy vegetable with a long history of cultivation. Clubroot is one of the most destructive diseases of cruciferous crops worldwide. In recent years, the incidence of clubroot disease has become increasingly severe, and the quality and yield of Chinese cabbage have been affected by it. Breeding clubroot-resistant (CR) varieties is the most effective and environmentally friendly control method. At present, several CR varieties with high-resistance properties have been developed, but their control effect of clubroot remains inadequate. An increasing number of resistance genes are demanded to aggregate in a single CR variety to meet the diversity of *P. brassicae* physiological races. Therefore, the pyramiding of multiple CR genes is of great significance to improve the resistance of these products to clubroot.

Clubroot is an obligate, parasitic, soil-borne disease caused by *P. brassicae*, which is commonly pathogenic to cruciferous crops [1,2]. Due to the long survival time of *P. brassicae* in soil, soil carrying dormant spores can easily lead to the persistence of the disease, causing severe damage that is difficult to control by physical, chemical, or biological practices [3,4,5]. If the plant itself is resistant to *P. brassicae*, the reliance on these controls would be reduced and more cost-effective. Consequently, disease resistance is an important trait in plant breeding, and cultivating resistant varieties using CR genes is the best way to control clubroot disease in Chinese cabbage [6]. At present, several CR-gene loci have been reported in *B. rapa*, namely, *Crr1*, *Crr2*, *Crr3*, *Crr4*, *CRa*, *CRb*, *CRc*, *CRd*, *CRk*, *PbBa3.1*, *PbBa3.2*, *PbBa8.1, Rcr1*, *Rcr4*, *Rcr8*, and *Rcr9* [4,7,8,9,10,11,12,13,14,15,16]. Among them, *Crr1, PbBa8.1*, and *Rcr9* were located in the A8 linkage group; *Crr2* in the A1 linkage group; *Crr4* in the A6 linkage group; *CRc* and *Rcr8* in the A2 linkage group; and *Crr3*, *CRa*, *CRb*, *CRd*, *CRk*, *Rcr2*, *Rcr4*, *PbBa3.1*, and *PbBa3.2* in the A3 linkage group. The identification of these CR loci and their corresponding linkage markers allowed us to pyramid multiple CR genes into a variety by marker-assisted selection (MAS), which could accelerate the screening process of resistance sources and the identification of resistance genes, additionally improving the efficiency of breeding selection.

Gene pyramiding is the process of aggregating two or more favorable genes through the marker-assisted-selection breeding pathway to develop excellent varieties [17]. The pyramiding of different target genes in the same crop can not only improve crop yield and quality, but also improve the agronomic traits and enhance resistance durability [18,19,20,21,22]. In terms of crop disease resistance, because of the differentiation of the physiological race of pathogens, varieties containing a single disease-resistance gene are no longer adequate enough to resist diseases and also lose their resistance in a short period of time, whereas pyramiding multiple disease-resistance genes can improve the durability of crop disease resistance and provide a greater economic value [23,24,25]. Watson and Singh first developed the process of gene pyramiding in wheat crops for rust resistance [26]. Hittalmani et al. developed rice lines carrying two or three blight-resistance genes: *Pi*1, *Pi*z5 and *Pi*ta. The disease-resistance test proved that the combination of resistance genes results in a greater resistance to leaf blight [27]. Pradhan et al. [28] conducted breeding work for rice bacterial blight resistance and eventually produced new varieties with pyramiding *Xa21*, *xa13,* and *xa5* genes, which are highly resistant to bacterial blight disease. In Brassica crops, Matsumoto et al. pyramided three CR genes of *CRa*, *CRc*, and *CRk* by combining MAS with conventional breeding in Chinese cabbages, and the results showed that the disease resistance of pyramided lines were well-improved [24]. Similarly, the pyramiding of a major and minor gene has been reported to increase the resistance of *Brassica napus* and *Brassica oleracea* to a variety of *P. brassicae* pathogenic types [29,30]. In our study, the *CRd* and *CRa* genes are pyramided in Chinese cabbage varieties by MAS combined with disease-resistance identification. Following multiple generations of backcrossing and self-crossing practices, novel varieties resistant to multiple *P. brassicae* pathotypes are generated in this study, which is of great significance for the development of CR-breeding practices for Chinese cabbages.

## 2. Materials and Methods

### 2.1. Plant and P. brassicae Materials

Two Chinese cabbage, clubroot-resistant (CR), inbred lines, CR252 and 85-74, were used for pyramiding resistance genes. CR252 is an inbred line with good commercial traits and possesses a resistance locus, *CRa*, which is used as a recurrent parent. The donor parent, 85-74, contains the *CRd* resistance locus [13].

Six field isolates of *P. brassicae*, namely, LNND-2, LNXM-1, SCXC-60, YNTC-75, LAB-5, and CQPL-99, were used for the resistance evaluation of the pyramiding lines, which have been classified as pathogens Pb3, Pb4, Pb5, Pb8, Pb9, and Pb12, according to the Sinitic clubroot differential set [31], respectively. The *P. brassicae* isolates were maintained on the roots of a susceptible Chinese cabbage variety, 91-12, and stored in a −40℃ refrigerator belonging to the Vegetable Molecular Biology Group, Shenyang Agricultural University.

### 2.2. Breeding Program for Gene Pyramiding

To pyramid *CRd* and *CRa* genes, crossing and selection for resistance characteristics were conducted from 2017 to 2020, as presented in Figure 1. The two paternal lines were crossed to produce the F_1_ hybrid, and then continuously backcrossed with CR252 as the recurrent parent to produce BC_4_F_1_ accompanied with marker-assisted selection (MAS) in each generation. Then, the selected BC_4_F_1_ lines were self-crossed to generate BC_4_F_2_. BC_4_F_2_ generation was screened for genotype identification, and homozygous lines with resistance-gene loci were then selected for the self-crossing process. Finally, homozygous, stable, BC_4_F_3_ lines with a high recovery rate of genetic background and high resistance to clubroot were obtained.

### 2.3. Development and Validation of Polymorphic Markers between Parents

Seven pairs of flanking SSR markers closely linked to the *CRd* gene and three markers based on the *CRa* gene sequence were selected for polymorphism screening as foreground markers between the parents (Appendix A). The DNA of the parents and lines of each generation was extracted by the modified CTAB (cetyltrimethylammonium bromide) method. The genomic DNA was amplified by PCR using a 10 μL reaction system containing 2 μL genomic DNA, 1 μL forward and reverse primers, 5 μL of 10× PCR buffer, and 2 μL of ddH_2_O. The PCR products of *CRa* gene-related markers were observed by using 1% agarose gel electrophoresis. A total of 6% polyacrylamide gel electrophoresis was used to detect the SSR markers designed for the *CRd* gene [13].

For the genetic background selection, a total of 1200 SSR primers covering 10 chromosomes of *B. rapa* were used to screen the polymorphisms between parents. Subsequently, the selected markers with good polymorphism properties, clear bands, and an even distribution on 10 chromosomes were used to screen the genomic background from the BC_3_F_1_ generation. The degree of backcross offspring recovery to recurrent parent was reflected by the value of the background recovery rate. The SSR marker information, PCR reaction, and amplification conditions used for background selection were referred to by Li et al. [32] and Ge et al. [33].

### 2.4. Investigation of the Phenotypic Characteristics

*P. brassicae* fluid was prepared for the clubroot disease-resistance identification of the parental lines, backcross generation, and pyramided lines. The swollen, infected roots were ground with a water solution, and then filtered through a gauze. The filtrate was collected into a 50 mL centrifuge tube, and the supernatant was discarded following repeated centrifugation. The precipitation was retained and dissolved in sterile water, and finally stored in a 4 °C refrigerator. The resting spores’ concentration was diluted to 1 × 10^7^ spores·mL^−1^. Inoculation was performed by injecting 2 mL of spore solution around the roots for each plant in a 72-well seeding tray at the stage when 3–4 true leaves had developed whilst in a greenhouse. A disease-resistance survey was conducted to record the incidence rate and grade at 45 days following *P. brassicae* inoculation. The disease symptoms were scored as follows: level 0: roots not visible, no clubs symbol, and plant growth normal; level 1; a few small clubs on lateral roots; level 2: larger clubs on lateral roots or small clubs on main roots; and level 3: main and lateral roots present significant swelling. The disease index (DI) value was calculated according to the formula DI = [(n_1_ + 2n_2_ + 3n_3_)/N_T_ × 3] × 100, where n_i_ represents the number of plants with the symptom of i and N_T_ is the total number of plants tested.

In the autumn of 2020 and 2021, two selected pyramided lines (BC_4_F_3_) were planted in a field in the Xinmin district of Shenyang, a clubroot-disease epidemic area, for field-resistance testing. The pathogen LNXM-1 collected in this field broke through the resistance of the *CRa* gene, but not the *CRd* locus.

The main agronomic traits of the pyramided lines and recurrent parent were investigated in the harvest period, including the outer-leaf length, outer-leaf width, outer-leaf number, outer-petiole length, outer-petiole width, petiole color, head weight, head length, head width, head shape, head solidity, head color, plant height, plant width, and plant weight (Figure 2). Twelve plants for each line were used for leaf and head related traits investigation. The pod traits were also investigated, including the pod length, pod width, number of seeds per pod, and hundred-grain weight. Moreover, the flowering time of the BC_4_F_2_ generation was measured, that is, the days from the seedling stage to the first flower’s emergence. An independent-samples t-test analysis was performed between pyramided line and recurrent parental line at *p* ≤ 0.05 significant level for all agronomic traits. If *p* value ≤ 0.05, the agronomic traits of the pyramided line would be considered to be significantly different from the recurrent parental line.

## 3. Results

### 3.1. Polymorphic-Marker Screening for Foreground Selection

Among the seven SSR markers designed for the *CRd* locus, yau106 and yau115 closely linked to *CRd* presented polymorphisms between CR252 and 85–74 (Figure 3). Marker CRa full-length, designed for cloning the full-length sequence of *CRa*, presented a polymorphism between two parental lines. *CRa* full-length gene sequence was only amplified in parental line CR252. These three markers were used for the foreground selection of each generation. The details of the markers are presented in Table 1.

### 3.2. Polymorphic Markers’ Screening for Background Selection

A total of 1200 genomic SSR markers designed based on the *B. rapa* genome sequence [32,33] were used for polymorphism screening between two parental lines. Finally, 95 polymorphic markers evenly distributed on 10 chromosomes were selected for background screening from the BC_3_F_1_ generation, of which the numbers of markers on chromosomes A1 to A10 were 13, 9, 11, 7, 9, 10, 10, 9, 9, and 8. The total physical length covered was 205.7 Mb. The average interval of markers was 2.6 Mb (Appendix A), and the distribution of these markers in the linkage groups is presented in Figure 4.

### 3.3. Foreground and Background Selection for Each Generation

Foreground selection was performed for each generation of backcross hybrids from BC_1_F_1_ to BC_4_F_1_ using the *CRa* and *CRd* linked markers previously screened. From the BC_3_F_1_ generation, genomic background selection was performed on the plants that met the selection results of the foreground marker.

For the foreground selection, twenty heterozygous lines were preliminarily screened with two *CRd* linkage markers, yau106 and yau115, and screened further with the *CRa* full-length marker out of 60 BC_1_F_1_ plants. Finally, two BC_1_F_1_ robust lines with two resistance loci were selected for further backcross. The 152 BC_2_F_1_ plants were screened by the same method, and 6 lines were used for further backcross.

In the BC_3_F_1_ generation, 76 plants that carried both *CRd* and *CRa* loci were screened out of 197 lines, with background recovery rates ranging from 69.47% to 85.26% (Figure 5). Among them, 10 lines with the highest recovery rate and good growth conditions were selected for self-crossing and backcrossing to obtain BC_3_F_2_and BC_4_F_1_. According to the results of the foreground marker screening, 26 lines with both *CRd* and *CRa* loci were obtained from 116 BC_4_F_1_ lines. Twenty-nine markers that had not recovered in recurrent parental genotypes in the BC_3_F_1_ generation were selected for the background selection of these 26 lines. The background recovery rates of 26 BC_4_F_1_ lines ranged from 78.95% to 97.89%, with one exceeding the expected value by 96.88% (Figure 5). Eight BC_4_F_1_ plants were selected for self-crossing to generate a BC_4_F_2_ population. Finally, six lines of BC_4_F_3_ were obtained from 11 BC_4_F_2_ lines with homozygous *CRa* and *CRd* loci and a 100% background recovery rate.

### 3.4. Clubroot-Resistance Identification and Morphological-Trait Evaluation of Pyramided Line

The clubroot-resistance identification of the parents and F_1_ generations was conducted by the inoculation of the *P. brassica* field isolate LNXM-1. The donor parent, ‘85-74′, was all-resistant to clubroot; the recipient parent, ‘CR252′, was all-susceptible; and 60 F1 (CR5274) plants showed all-resistance traits (Table 2). 

Indoor disease-resistance identification was also performed for BC_1_F_1_ and BC_4_F_2_ generations. The chi-squared tests conducted for BC_1_F_1_ and BC_4_F_2_ generations were consistent with the Mendelian genetic segregation ratios of 1:1 and 3:1, respectively (Table 3). To better verify the resistance of the backcross progeny to the LNXM-1 pathotype, the BC_4_F_2_ generation and five pyramided homologous BC_4_F_3_ lines (CR5274①-4, CR5274⑥-2, CR5274⑥-4, CR5274⑥-8 and CR5274⑩-4) were planted in the infected field in the Xinmin area of Shenyang. The separation ratio of resistant to susceptible plants of 67 BC_4_F_2_ was 3:1, according to the chi-squared test (χ^2^ = 0.12) (Table 4). All five gene-pyramided BC_4_F_3_ lines presented strong resistance in the infected field (Table 4).

To further validate the disease resistance of gene-pyramiding lines to different physiological races, the lines of 85-74, CR252, CR5274⑥-4, and CR5274⑥-8 were selected for inoculation. The parent CR252 carrying the *CRa* gene was resistant to Pb3, but susceptible to Pb4, Pb5, Pb8, and Pb11. Parental line 85–74 harboring the *CRd* gene was resistant to Pb4, Pb5, Pb8, and Pb11, but susceptible to Pb3. Compared to the two parental lines carrying a single disease-resistance gene, the gene-pyramided lines were resistant to multiple physiological races (Table 5).

The agronomic traits of the parents and two pyramided lines, CR5274⑥-4 and CR5274⑥-8, were also investigated, including the growth period and 16 heading-related agronomical characters. In addition, eight lines of the BC_4_F_2_ generation and the recurrent parent CR252 were used for the evaluation of the pod characters. The results show that there are no differences in any of the heading-related agronomic traits and pod-related traits between the progenies and recurrent parent, which confirms that the backcross progenies restore the genome of the recurrent parent. The agronomical and pod characters of the progeny and recurrent parents are presented in Table 6 and Table 7.

## 4. Discussion

The marker-assisted backcross breeding that introgresses two or more genes for targeted traits into the elite cultivars has been widely used in several crops. The study showed that plants pyramided with multiple resistance genes improved their resistance properties compared to the parents, and the agronomic traits and qualities were equivalent or better than those of the parents, such as rice [34,35,36,37], soybean [38,39], and tomato [40]. A representative example for rice is the breeding of rice varieties resistant to bacterial blight (BB) and sheath blight disease. At present, lines introgressed with several BB-resistance genes (xa4, xa5, xa7, xa13, xa21 have been successfully cultivated using MAS and exhibit greater resistance than the lines with only one or two genes [36,37,41,42]. Ramalingam et al. (2020) improved two rice cultivars through introgression of three BB-resistance, one blast resistance, and three sheath blight resistance genes/QTLs. The pyramided lines of rice exhibited a high degree of resistance to these three diseases [36]. Compared to the field crop such as rice, wheat and maize, there are fewer reports on gene pyramiding in Chinese cabbage. To date, several CR genes/QTL have been identified from diverse Brassica resistant sources, of which *CRa*, *CRb*, and *PbBa8.1* are well utilized in clubroot resistance breeding programs. In Brassica crops, most of the CR cultivars of *B. rapa* and *B. napus* harboring a single CR gene can easily be broken down by the new *P. brassicae* pathotypes. *CRb* and *PbBa8.1* are two resistance genes genetically identified from *B. rapa*, and pyramided in *B. napus* through MAS [29,43]. The *CRb* and *PbBa8.1* pyramided *B. napus* line exhibited high levels of resistance to most *P. brassicae* isolates than that of a single CR gene [29]; however, the pyramided line was susceptible to an isolate collected from the Xinmin area in Liaoning province, which was identified as Pb4 by the SCD system [31]. Tomita et al. (2013) accumulated multiple CR-QTLs in *B. oleracea* conferred broad-spectrum clubroot resistance against six *P. brassicae* isolates [30]. In Chinese cabbage, Matsumoto et al. (2012) pyramided three clubroot resistant genes (*CRa*, *CRk*, and *CRc*) through MAS to improve the resistance. The *CRd* was a new locus identified from a Chinese cabbage line, 85-74, and presented great resistance to Pb4, but was susceptible to Pb3 [31], whereas the CR252 parental line was resistant to Pb3. Following the pyramiding process, the pyramided lines exhibited great resistance to multiple *P. brassicae* isolates. In the future, the improved pyramided lines could be tested in multiple clubroot infected areas and used as a potential donor in hybridization breeding to develop clubroot resistant cultivars in Chinese cabbage. In addition, we previously developed a pair of *CRb* near isogenic lines and used for transcriptome analysi to reveal the molecular mechanism of *CRb* gene of *B. rapa* against *P. brassicae* infection [44]. Thus, the new generated *CRa* and *CRd* gene pyramided homozygouse lines could be also utilized for multi-omics analysis and gene interaction studies.

Background selection is performed to feed back the extent to which the backcrossing progeny is recovered to the recurrent genome. Individuals with the highest ratio of recurrent parent genomes in each backcross generation were selected by marker-assisted selection as the parent of the subsequent backcross, and the recurrent parent genomes could be completely recovered within three backcross cycles, while conventional breeding required six generations [45]. In our study, a total of 95 SSR markers generated polymorphisms between the parental genotypes, ‘CR252′ and ‘85-74′, which were used to estimate the recovery rates of the recurrent parental genome in the pyramided BC_3_F_1_ genotypes. Chen et al. [46] observed that when selecting the genomes of recurrent parents, following three backcross cycles, approximately 98.8% of the genetic background information was obtained from recurrent parents. However, in this experiment, the background-marker selection in the BC_3_F_1_ generation of ‘CR5274′ did not achieve the desired recovery rate, and only plants that exceeded the expected recovery rate appeared in the BC_4_ generation. In general, our expectations were not met. This may have been caused by the fact that we used a small population of plants, performed late background-screening generations, or the presence of linkage drag. If the screening of background markers was performed at a later stage of breeding, the population size was necessary, and the number of populations needed to be expanded to meet the requirements for breeding. In addition, the linkage drag occurring during backcross-breeding process can lead to the introduction of other linkage genes in addition to the target gene, thus affecting the final effect [47].

In conclusion, two clubroot-resistant genes, *CRd* and *CRa*, were pyramided into Chinese cabbage samples by MAS, and the new Chinese cabbage pyramided material presenting excellent agronomic characters was obtained by the backcross-breeding method. The cultivated pyramided line recovered the phenotypic traits of the recurrent parents and strengthened clubroot resistance in *B. rapa*. The present study is highly significant for the development of breeding practices to develop a greater resistance to clubroot for Chinese cabbages.

## Figures and Tables

**Figure 1 genes-13-02414-f001:**
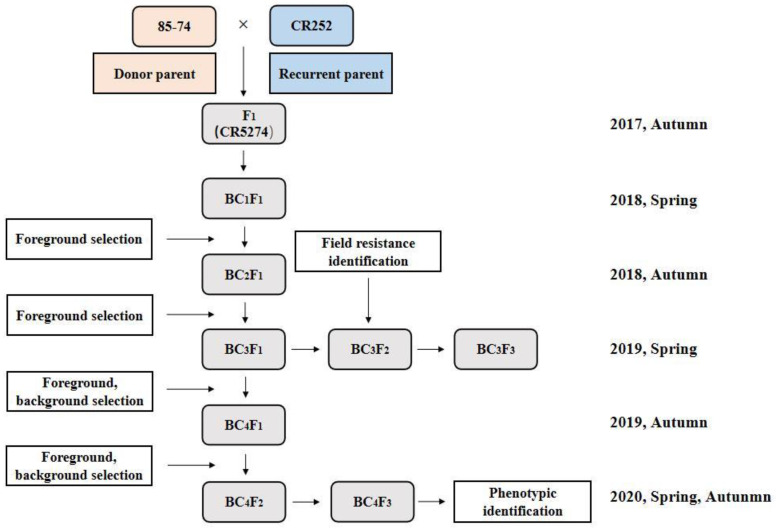
Schematic diagram of marker-assisted gene pyramiding. CR252: recurrent parent; 85-74: donor parent.

**Figure 2 genes-13-02414-f002:**
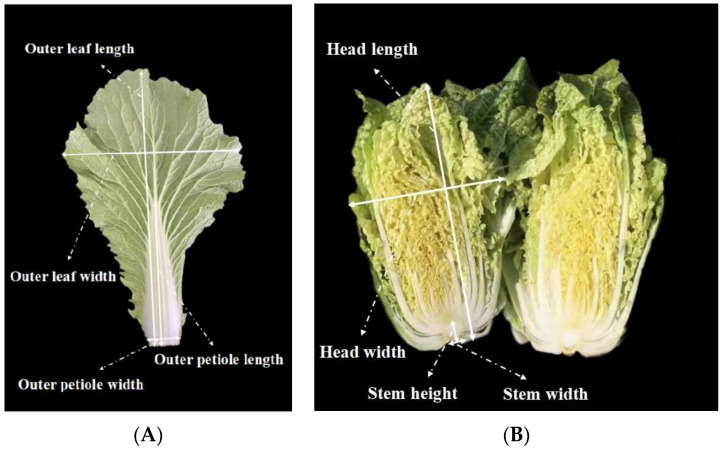
(**A**) The measurement standard of outer-leaf length, outer-leaf width, outer-petiole length, and outer-petiole width. (**B**) The measurement standard of head length, head width, Stem height, and Stem width.

**Figure 3 genes-13-02414-f003:**
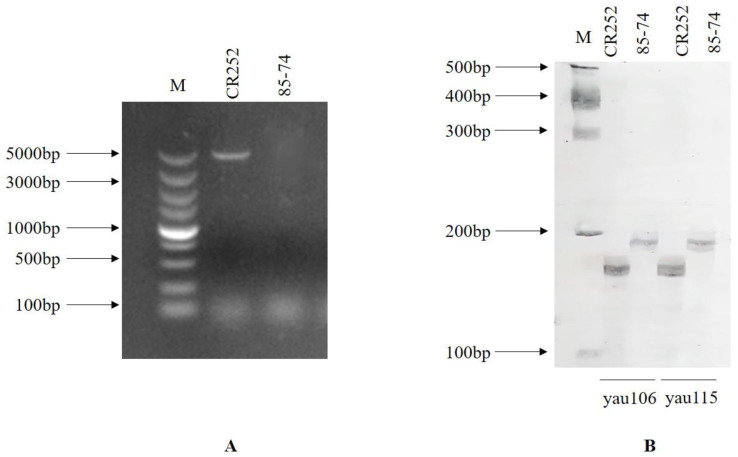
Screening results for polymorphic markers for foreground selection. (**A**) polymorphism of *CRa* full-length marker between CR252 and 85–74. (**B**) polymorphism of *CRd* linked markers yau106 and yau115 between CR252 and 85–74, respectively.

**Figure 4 genes-13-02414-f004:**
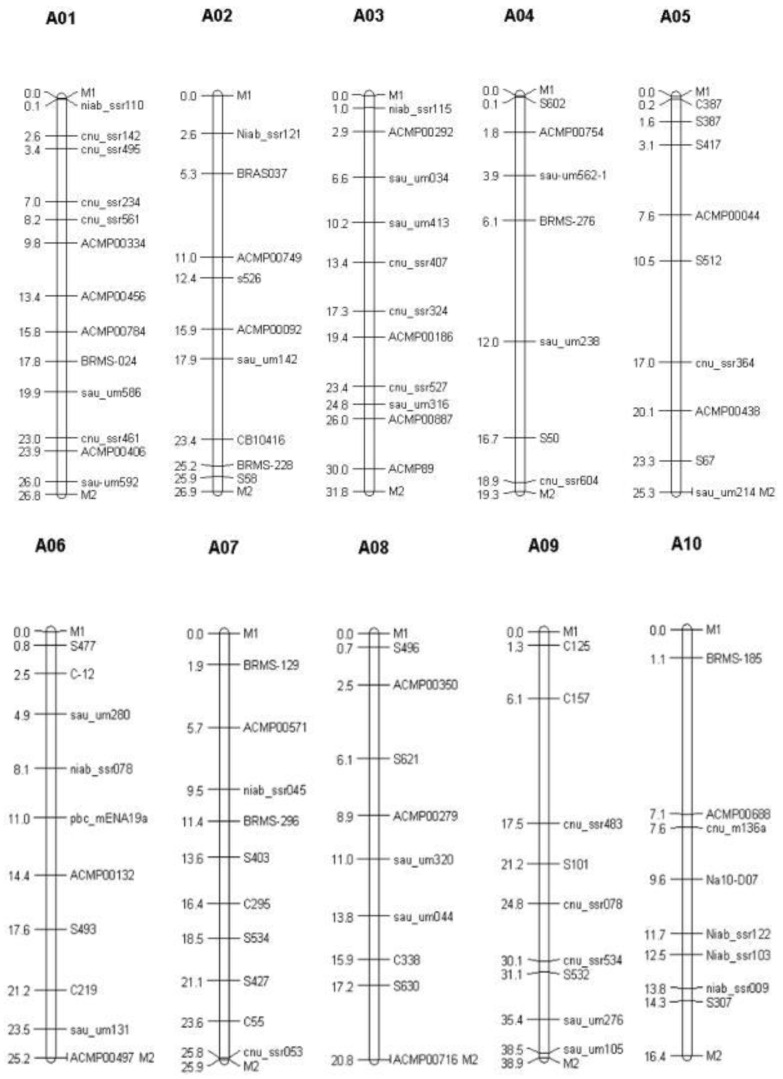
Distribution of background-selection markers in *B. rapa* chromosomes.

**Figure 5 genes-13-02414-f005:**
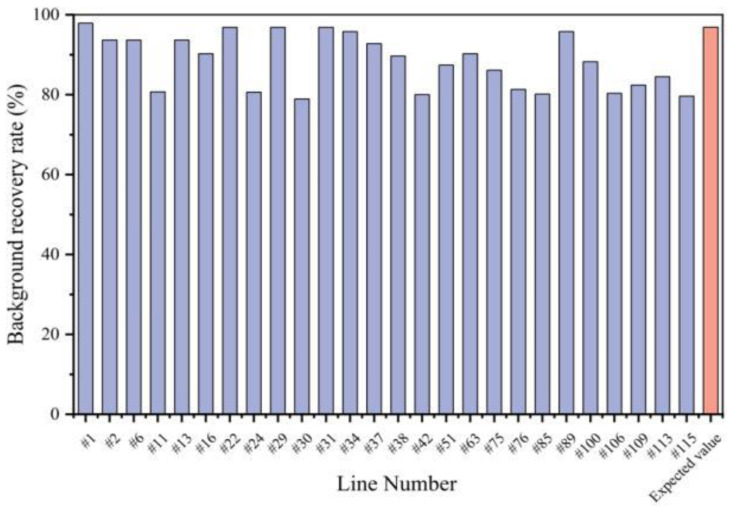
Background recovery rate of 26 BC_4_F_1_ individual plants (#1~#115).

**Table 1 genes-13-02414-t001:** Molecular marker details for foreground selection of two CR genes.

Markers	CR Gene	Chromosome	Primer Sequence
yau106	*CRd*	A3	F: GGTCACCAATCGAAGCCTTT R: GCATGCGGGTATACACATCT
yau115	*CRd*	A3	F: CCGTTTGGTTTCCCTTGCAA R: GTTTAACACAGCAACAACAATGG
*CRa* full-length	*CRa*	A3	F: ATGGATTTCTCTCTTTTCC R: TTAACATGAGGGAGTTTCCAG

**Table 2 genes-13-02414-t002:** Disease-resistance verification results of parental lines and F_1_ generation by inoculation of LNXM-1 *P. brassicae* isolate.

	Total	Level 0	Level 1	Level 2	Level 3	Incidence Index
85–74	34	34				**0**
CR252	30				30	100
F_1_ (CR5274)	60	60				0

**Table 3 genes-13-02414-t003:** Disease resistance of various generations of CR5274.

	Total	Level 0	Level 1	Level 2	Level 3	Resistance: Susceptibility	Theoretical Ratio	χ^2^
CR5274BC_1_F_1_	60	28	6		26	28:32	1:1	0.27
CR5274F(BC_4_F_2_)	165	116	4	8	37	116:49	3:1	0.49

**Table 4 genes-13-02414-t004:** Disease statistics of different populations in the field.

	Total	Level 0	Level 1	Level 2	Level 3	Resistance: Susceptibility	Theoretical Ratio	χ^2^
CR252	8				8	-	-	-
85-74	8	8						
CR5274F(BC_4_F_2_) *	67	39	10	3	15	49:18	3:1	0.12
CR5274①-4	15	15						
CR5274⑥-2	15	15						
CR5274⑥-4	14	14						
CR5274⑥-8	15	15						
CR5274⑩-4	15	15						

* As the level 1 galls identified in the field are very small, the identified plants were grown normally and are not affected for harvest, so the plants with level 1 galls are classified as disease resistant plants in this field identification trial.

**Table 5 genes-13-02414-t005:** Resistance identification of gene-pyramided lines inoculated with different *P. brassicae* races.

Materials	Level	85–74	CR252	CR5274⑥-4	CR5274⑥-8
LNND-2 (Pb3)	0		28	30	29
(Pb3)	1				
	2				
	3	23			
CQPL-99 (Pb4)	0	18		29	30
(Pb12)	1				
	2		3		
	3		20		
SCXC-60 (Pb5)	0	19		27	22
(Pb5)	1				
	2				
	3		16		
YNTC-75 (Pb8)	0	15		28	26
(Pb8)	1				
	2				
	3		22		
LAB-5 (Pb11)	0	20		26	30
(Pb9)	1				
	2				
	3		17		

Note: LNND-2 (Pb3), CQPL-99 (Pb12), SCXC-60 (Pb5), YNTC-75 (Pb8), and LAB-5 (Pb9) are the *P. brassicae* races inoculated for resistance identification.

**Table 6 genes-13-02414-t006:** Agronomic-trait investigation of recurrent parent and pyramided lines.

Agronomic Traits	CR252	CR5274⑥-4	CR5274⑥-8
Plant height (cm)	34.00 ± 2.64	33.33 ± 2.08	31.00 ± 1.73
Plant width (cm)	49.33 ± 2.30	48.00 ± 2.64	44.33 ± 2.04
Plant weight (g)	2.07 ± 0.18	2.35 ± 0.17	2.07 ± 0.18
Number of outer leaves	9.00 ± 1.00	9.00 ± 3.00	9.66 ± 1.52
Leaf length (cm)	32.83 ± 2.51	33.66 ± 2.84	32.00 ± 2.64
Petiole length (cm)	18.50 ± 1.50	19.83 ± 1.25	19.50 ± 3.50
Leaf width (cm)	19.33 ± 2.88	19.33 ± 1.52	20.00 ± 2.00
Petiole width (cm)	5.30 ± 0.57	5.83 ± 0.28	6.00 ± 0.50
Petiole color	Green	Green	Green
Head weight (g)	1.12 ± 0.08	1.18 ± 0.01	1.14 ± 0.141
Head shape	Folded	Folded	Folded
Head solidity	Compaction	Compaction	Compaction
Head length (cm)	23.30 ± 1.13	23.33 ± 1.52	23.66 ± 1.52
Head width (cm)	11.50 ± 1.32	12.00 ± 1.00	12.66 ± 0.57
Stem height (cm)	3.50 ± 0.50	4.5 ± 1.32	3.00 ± 0.00
Stem width (cm)	2.93 ± 0.11	3.06 ± 0.11	2.76 ± 0.40
Head color	Light yellow	Light yellow	Light yellow
Growth period (day)	80d ± 0.00	80d ± 0.00	80d ± 0.00

Note: Significance level for *t*-test was at *p* ≤ 0.05. No asterisk in the table represented no significant differences between pyramided lines and recurrent parental line for leaf and head related traits at *p* ≤ 0.05 level.

**Table 7 genes-13-02414-t007:** Differences in main, botanical characters of offspring and recurrent parent.

	Pod Length (cm)	Pod Width (cm)	Number of Seeds per Pod	Hundred-Grain Weight (g)
CR5274-1	5.20 ± 0.26	0.50 ± 0.00	12.00 ± 2.00	0.34 ± 0.00
CR5274-2	5.10 ± 0.20	0.53 ± 0.05	11.33 ± 1.52	0.34 ± 0.11
CR5274-3	5.13 ± 0.11	0.60 ± 0.00	12.66 ± 0.57	0.35 ± 0.11
CR5274-6	5.10 ± 0.26	0.53 ± 0.05	11.66 ± 2.30	0.35 ± 0.15
CR5274-22	5.00 ± 0.20	0.53 ± 0.05	12.33 ± 2.51	0.35 ± 0.26
CR5274-29	5.03 ± 0.15	0.50 ± 0.00	11.00 ± 1.00	0.34 ± 0.10
CR5274-31	5.03 ± 0.05	0.60 ± 0.00	12.33 ± 1.52	0.35 ± 0.10
CR5274-34	5.13 ± 0.15	0.53 ± 0.05	12.00 ± 2.00	0.35 ± 0.15
CR252	5.10 ± 0.30	0.5 ± 0.05	12.00 ± 2.00	0.34 ± 0.20

Note: CR5274-1, CR5274-2, CR5274-3, CR5274-6, CR5274-22, CR5274-29, CR5274-31, and CR5274-34 are lines of the BC_4_F_2_ generation. Significance level for *t*-test was at *p* ≤ 0.05. No asterisk in the table represented no significant differences between pyramided lines and recurrent parental line for pod-trait at *p* ≤ 0.05 level.

## Data Availability

The data presented in this study are available upon requestfrom the corresponding author.

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
