# Peer review of "Marker-Assisted Pyramiding of CRa and CRd Genes to Improve the Clubroot Resistance of Brassica rapa"

_genes, 2022, doi:10.3390/genes13122414_

Round 1

Reviewer 1 Report

The manuscript (MS) submitted by Li et al. is focused on the pyramiding of CRa and CRd genes in Chinese cabbage using MAS and finally authors obtained new homozygous pyramided lines. I applaud the volume of work done by the author's group. The MS is well written, and the information produced by this project is valuable. The experiment was well designed and performed. It is considered to be published in journal ‘Genes’. Below are some specific comments, although not all issues are addressed

· Authors’ carried out the analysis of genetic background recovery from BC3F1 onward, why they were not performing background selection from BC1F1 generation?

· Page 6: [For foreground selection, twenty robust lines in BC1F1 generation were selected] – What do you mean by twenty robust lines? these are number of plants or other? Clarify? If plants, then the population size is too less for screening.

· Number of plants screened at initial backcross generations was too less?

English spell check required

Reviewer 2 Report

Marker-assisted pyramiding of CRa and CRd genes to improve clubroot resistance of Brassica rapa is good attemp with a scientific approach. Figure 3 is not much clear.

This study provide new CR gene pyramided lines for the development of clubroot resistance Brassica varieties in future breeding programs.

Paper is well written

. I would suggest to recheck the results of parental lines and F1 generation by inoculation of LNXM-1 P. brassicae isolate.

Some references are not complete.

I am not satisfied with discussion section.

Improve discussion with international research with up to date references. There are some minor grammatical mistakes in the text.

Revisions are needed
